# The Trade-Offs and Synergistic Relationships between Grassland Ecosystem Functions in the Yellow River Basin

**Jie Yang [1], Baopeng Xie [2],\* and Degang Zhang [1]**

[1] School of Pratacultural Science, Gansu Agricultural University, Lanzhou 730070, China; 18293893322@163.com (J.Y.); zhangdg@gsau.edu.cn (D.Z.)

[2] School of Management, Gansu Agricultural University, Lanzhou 730070, China

\* Correspondence: xiebp@gsau.edu.cn

**Abstract:** Grassland is the most important land use/cover type in the Yellow River basin. Studying its ecosystem services and the trade-off and synergistic relationships between its various functions is of great significance to high-quality development and the protection of the ecological environment in the Yellow River basin. This paper evaluates the five typical functions of grassland in the Yellow River basin quantitatively, including water yield, carbon storage, soil conservation, habitat quality, and NPP by adopting the InVEST model and the CASA model. It analyzes changes in the trade-offs and the synergistic relationships between the five ecosystem functions from 1990 to 2018 by adopting the correlation coefficient method. The paper also analyzes and explores the spatial heterogeneity of the trade-offs and synergistic relationships by adopting the bivariate spatial autocorrelation method. The results show that from 1990 to 2018, the average water yield depth, carbon storage, and NPP of the grassland in the Yellow River basin tended to increase; soil conservation and habitat quality showed a decreasing trend; and the spatial distribution of the five functions were clearly in line with zonal law. The five ecological functions were synergistic; the synergistic relationship between water yield and the other functions was relatively weak, and there was a strong synergistic relationship between the other four functions. The trade-offs and synergistic relationships between the five functions demonstrated significant spatial heterogeneity in space. This research provides a scientific basis for determining the optimal utilization and sustainable development of grassland resources.

**Keywords:** ecosystem service of grassland; trade-off and synergistic relationship; InVEST model; Yellow River basin

## 1. Introduction

Grassland is the largest terrestrial ecosystem in China, performing multiple functions, such as animal husbandry production, biodiversity, soil and water conservation, and carbon storage, and making significant contributions to socio-economic development and ecosystem balance [1,2]. Due to climate change and human interference, about 50% of the grassland in the world is degrading [3], resulting in an overall decline in grassland ecosystem functions, causing problems such as reduced biomass, reduced biodiversity, and increased soil erosion [4,5]. The major cause of the above problems is the imbalance between the various ecological functions of grassland. For example, although overgrazing has played a more significant role than animal husbandry in the production of grassland, it destroys the restoration of grassland vegetation while also affecting the performance of other ecological functions. Therefore, coordinating the relationship between the various ecological functions of grassland is essential for the rational allocation of grassland production and ecological functions and the improvement of the overall functions of grassland ecological services; it is also conducive to guiding humans to use grassland resources rationally. In order to do this, it is essential to clarify the relationship and the mechanism of action and changes between the various functions. The correct understanding of the

relationship between the ecosystem functions is the basis of decision-making for the sustainable development of regional ecosystem services [6]. Studies have found that, due to the diversity of grassland ecosystems and the complexity of human selection, overemphasizing certain ecosystem functions reduces the impact of other ecosystem functions. With changes in resources and environmental conditions and human utilization, the trade-off and synergy between ecosystem services will undergo mutual conversion. The spatial heterogeneity and time dependence of the trade-off and synergy have been verified by relevant studies [7]. This has promoted research on the interaction between ecosystem functions and trade-off management decision-making.

Scholars have carried out related research on grassland ecosystem services. For example, Lu et al.'s study on grassland in the Loess Plateau found that during 2000–2008, with the implementation of ecological restoration policies, soil conservation, carbon balance, and biomass and other ecological functions all increased [8]; Wu et al. [9], Liu et al. [10], and Zhang et al. [11] estimated the water conservation of a grassland ecosystem and discussed its temporal and spatial characteristics, adopting the precipitation storage method and the InVEST model; Zhong Juntao et al., selected three key ecosystem functions, namely carbon storage, soil conservation, and water conservation, to assess ecosystem functions before and after the grazing prohibition in Yanchi County, Ningxia province, in the interlaced area of agriculture and animal husbandry (2000, 2015) [12]; Lv studied the impact of grazing tourism compound pressure on the grassland ecosystem functions of Yulong Snow Mountain [13]. Byrd et al., explored the influence factors of climate change and human activities on a grassland ecosystem [14]. Du et al., found that under light grazing intensity, the above-ground biomass did not decrease, but the water conservation capacity showed a downward trend [15,16]. With the increasing grazing intensity and grassland degradation, the above-ground biomass and water conservation capacity showed a downward trend [17].

In recent years, research on the trade-offs and synergy of environmental functions has proliferated at an exponential rate. Mouchet et al., reviewed and analyzed the main methods of exploring trade-off relationships quantitatively in existing research [18]. Deng et al., summarized the main tools and processes applied in the analysis of trade-off relationships in land use management research [19]. Kragt et al., analyzed the trade-off relationship between Western Australia's agricultural products and ecosystem functions based on the APSIM model, and further applied the research on ecosystem functions to agricultural production [20]. Su et al. constructed the Human Activity Index (HMI), taking the Yanhe basin of the Loess Plateau as the research object and taking the township as the spatial scale. On this basis, they analyzed the trade-offs and synergistic relationship between index and ecosystem functions such as NPP, carbon sink, water yield, and soil and water conservation [21]. Haase et al., conducted an integrated study on the interrelationships of five ecosystem functions in urbanized areas in Germany, using spatial mapping methods. The results showed that due to changes in land use, there was a synergistic relationship between biodiversity and climate regulation functions; however, there was a trade-off relationship between climate regulation and cultural entertainment functions [22]. Thompson et al., conducted scenario simulations of four types of land use from 2010 to 2060, and characterized the correlation between ecosystem functions, taking Massachusetts as the research object, based on the intuitive spatial landscape model. The results showed that an increase in human activity possibly led to a decrease in the supply services of the ecosystem [23]. Using the extreme value method, Rao Sheng et al., performed a trade-off analysis on the use of grassland ecosystem services in Zhenglan Bannerused, taking biomass as the key variable of ecosystem service value [24]. Li et al., and Yang et al., studied the trade-offs and synergy of their different ecosystem functions, taking the Guanzhong-Tianshui Economic Zone as the research object and using correlation coefficients, spatial mapping, rose diagrams, and production possibility boundary methods [25,26]. The above-mentioned studies mostly focused on exploring ecosystem services on the scale of administrative divisions and the overall scale. At the same time, the analysis of trade-offs and synergy was based on the quantitative analysis of statistical relationships to reflect the overall

difference of the respective regions. Studies on the expression of temporal and spatial differences within the region, on the systematic understanding of the internal mechanism of the formation of ecosystem functions relationships, and on the internal heterogeneity of natural ecosystems are lacking.

The Yellow River basin is an important ecological barrier and economic zone in China [26]. Grassland is the main land use type and an important part of the natural ecosystem in the basin, which accounts for about 50% of the total area of the basin and plays important ecological roles, such as sand fixation, soil conservation, climate regulation, air purification, and water conservation. Due to frequent human activity and economic development in the Yellow River basin, the impact on the grassland ecosystem has gradually increased, and the retrograde succession of grassland has accelerated, leading to a series of ecological problems. The annual runoff of the Yellow River basin is mainly supplied by precipitation. Due to the influence of the atmospheric environment, the low rainfall and strong evapotranspiration capacity result in a very low water yield coefficient. The loose soil and the large slope in the middle reaches of the Yellow River basin have caused serious soil erosion. In recent years, the acceleration of urbanization and economic development have led to a decrease in biodiversity in downstream areas. At the same time, there are a large amount of grasslands and woodlands, which are important carbon sources, in the Yellow River basin. Ever-increasing human activity and climate change in the Yellow River basin have jointly led to changes in the relationship between the grassland ecosystem and its functions. Until now, research on the grassland ecosystem in the Yellow River basin has mostly focused on the status and mechanism of degradation [27], the classification of grassland desertification [28] and the evolution of the grassland's landscape [29]. Studies of the interrelationships between ecosystem functions are lacking. Based on this, this study took the grassland ecosystem of the Yellow River basin as the research object and adopted InVEST (Integrated Valuation of Ecosystem Services and Trade Offs) and CASA (Carnegie-Ames-Stanford) models to evaluate the following five functions: water yield, soil conservation, carbon storage, habitat quality and net primary productivity (NPP). We explored the relationships between the functions, using the correlation analysis method. Combing with the GeoDa spatial bivariate analysis method, we spatially expressed the trade-offs and synergistic relationships, showing the impact of the Yellow River basin's multi-dimensional zonality on the ecosystem function relationships from both macro and micro perspectives. Furthermore, the study highlighted the relevant changes between the ecosystem functions using decomposition and integrated perspective. This research can help decision-makers to formulate corresponding management strategies at appropriate spatial scales, and at the same time provide a basic reference for the ecological protection of mountains, waters, forests, fields, lakes, and grasses between regions.

## 2. Study Area

The Yellow River is the second largest river in China. It is located at a 96–119° east longitude and a 32–42° north latitude (Figure 1). It originates in the Bayan Har Mountains of the Qinghai-Tibet Plateau and flows into the Bohai Sea in Kenli County, Shandong Province. The main stream has a total length of 5464 km, a length of 1900 km from east to west, and a width of 1100 km from north to south. The total area of the basin is $79.5 \times 10^4$ km$^2$ (including the internal flow area of $4.2 \times 10^4$ km$^2$). The basin is vast, traversing the Qinghai-Tibet Plateau, the Inner Mongolia Plateau, the Loess Plateau, and the Huanghuaihai Plain from west to east. The topography of the basin is high in the west and low in the east, with a drop of 4480 m. The river source area in the west has an average elevation of more than 4000 m. There are many mountains and much snow throughout the year. The area is rich in precipitation and is an important water-producing area in the Yellow River basin. At the same time, the slope is large, the terrain is undulating, and the vegetation is seriously degraded. The central area is between 1000–2000 m above sea level, the geological structure of this area is broken, its soil texture is loose, its soil erosion is serious, and its soil conservation is weak. The eastern part, which is mainly composed of

the alluvial plains of the Yellow River, is mostly no more than 50 m above sea level and is mostly distributed among forest lands. It is an important carbon storage area and economic belt in the Yellow River basin. The expansion of construction land and a large degree of disturbance by human activity have resulted in poor-quality habitats and low biodiversity.

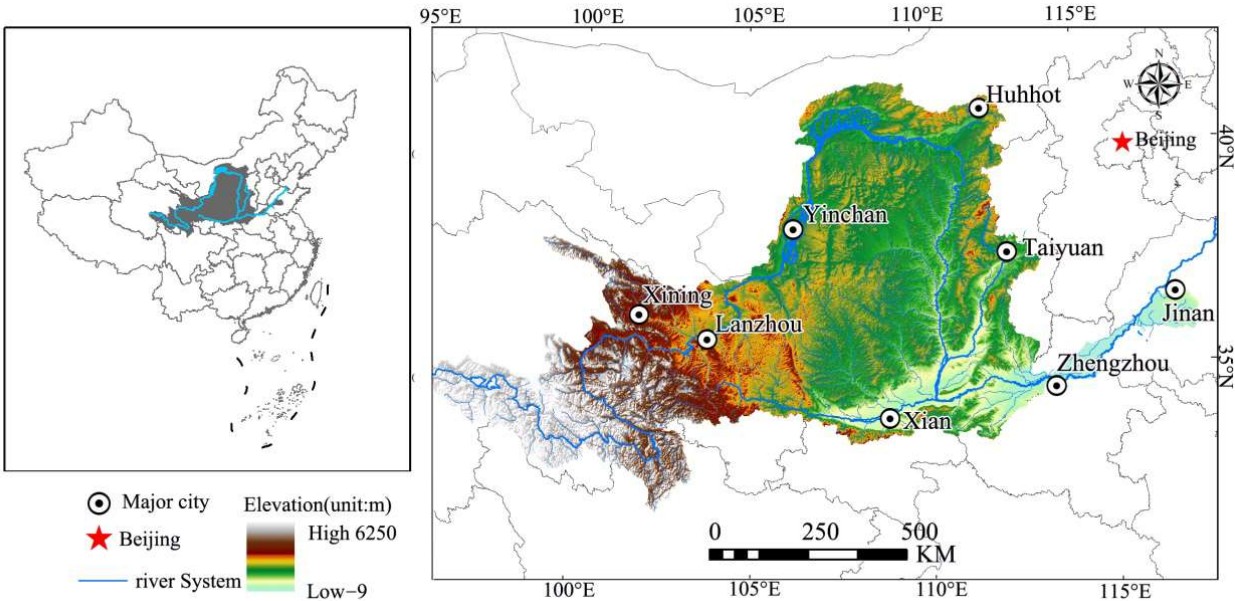

**Figure 1.** Range of Yellow River basin and distribution of river system.

## 3. Data and Methodology

### 3.1. Data Sources

This study used multi-source data sets, including land use/cover data sets, satellite imagery data sets, meteorological data sets, soil data sets and related auxiliary data sets. Specifically, the land use/cover data, whose spatial resolution is 1 km × 1 km, came from the national geographical conditions monitoring cloud platform (http://www.dsac.cn, accessed on 28 November 2020). The land use types were divided into eight categories: cultivated land, forest land, high-coverage grassland, medium-coverage grassland, low-coverage grassland, water area, construction land, and unused land. Data such as elevation (DEM), GDP and population density, with a resolution of 1 km × 1 km, were from the Chinese Academy of Sciences Resources and Environment Science and Data Center (http://www.resdc.cn, accessed on 28 November 2020). The monthly precipitation data came from the National Qinghai-Tibet Plateau Data Center (https://data.tpdc.ac.cn, accessed on 28 November 2020), China's monthly precipitation data set (doi:10.5281/zenodo.3185722), with a spatial resolution of 1 km × 1 km, using monthly data to obtain annual precipitation data through grid summation. The soil, sand, silt, clay, organic carbon content, maximum soil root depth and other data came from the National Glacier, Frozen Soil and Desert Science Data Center (shttp://www.ncdc.ac.cn, accessed on 28 November 2020)' 1:1 million soil database, with a spatial resolution of 1 km × 1 km. The top layer radiation data of the sun and atmosphere came from the global potential evapotranspiration and global drought index data set (https://cgiarcsi.community/data/global-aridity-and-pet-database, accessed on 28 November 2020), with a resolution of 1 km × 1 km. The NDVI data came from the United States Geological Survey (USGS) https://www.usgs.gov, accessed on 28 November 2020; the maxmum NDVI value of each pixel as calculated using the maximum value synthesis (MVC) method, and the spatial resolution was resampled to 1 km × 1 km. Basic geographic data such as roads, rivers, and the administrative boundaries of districts and counties came from the National Geographic Information Center.

*3.2. Method*

3.2.1. Water Yield

The water yield module in the InVEST model was adopted to evaluate the water yield of the grassland. This module is based on the principle of water balance, namely, the precipitation of each grid unit minus the actual evapotranspiration to get the water yield of the unit [30]. The calculation formula is as follows [31]:

$$Y_{xj} = P_x - AET_{xj} = \left(1 - \frac{AET_{xj}}{P_x}\right) \times P_x$$

In the formula, $Y_{xj}$ is the water yield of the $j$-type land use type grid $x$; $P_x$ is the actual annual precipitation of the grid $x$; and $AET_{xj}$ is the actual evapotranspiration of the $j$ land-use type grid $x$.

$$\frac{AET_{xj}}{P_x} = \frac{1 + w_x R_x}{1 + w_x R_x + 1/R_x}$$

In the formula, $R_x$ is the aridity index of grid $x$, which can be calculated from the potential evapotranspiration and rainfall; $\omega_x$ is an empirical parameter, dimensionless, used to describe climate-soil properties, and can be calculated from the available water content of vegetation and annual rainfall.

3.2.2. Soil Conservation

The InVEST model sedimentation module was used to calculate soil conservation. The model adopts the soil circulation equation to predict the actual soil erosion of each pixel in the watershed [32]: $USLE = R \times K \times LS \times P \times C$. In the formula, $USLE$ is the average annual soil erosion, $R$ is the rainfall erosion factor, $K$ is the soil erosivity factor, $LS$ is the slope and length factor, $C$ is the vegetation and management factor, and $P$ is the factor supporting protection measures. The model does not consider the $CP$ factor to calculate the potential erosion amount of the original land cover, assumes that the landscape is bare [32], and obtains the potential soil erosion amount: $RKLS = R \times K \times LS$. Based on the two soil erosion amounts, the soil conservation amount is obtained as: $SC = RKLS - USLE$. $SC$ is the amount of soil conservation.

3.2.3. Carbon Storage

Using the InVEST model's carbon storage module assessment, the module uses the land use map and the carbon density of each land use type to estimate the carbon storage of each unit. By evaluating the carbon storage in different periods, it can reflect the carbon storage function of the landscape. The calculation formula is as follows:

$$C_{(x,y)} = A_{(x,y)} \times \left(D_{(x,y)}^A + D_{(x,y)}^B\right) + D_{(x,y)}^S + D_{(x,y)}^D$$

In the formula, $C(x,y)$ represents the carbon storage of $y$ land use type $x$ unit; $A\ (x,y)$ represents the area of the unit $(x,y)$; and $D_{(x,y)}^A$, $D_{(x,y)}^B$, $D_{(x,y)}^S$, $D_{(x,y)}^D$ represent $y$ land use type $x$ unit above ground, underground, soil organic matter and dead organic carbon density, respectively.

3.2.4. Habitat Quality

The habitat quality module in the InVEST model was used for the assessment. By establishing the relationship between the land use data and threat factors, and comprehensively considering the influence, distance and intensity of the threat factors, it calculates the negative impact of threat factors on the habitat, and obtains the degradation of the habitat. The degree of degradation and habitat suitability are then used to calculate the

habitat quality to reflect the level of regional biodiversity [33]. The calculation formula is in the following form:

$$Q_{xj} = H_j \left( 1 - \left( \frac{D_{xj}^z}{D_{xj}^z + k^z} \right) \right)$$

In the formula, $Q_{xj}$ is the habitat quality index of grid $x$ in land use $j$; $H_j$ is the habitat suitability of habitat type $j$, with the range of [0, 1]; $D_{xj}$ is the habitat degradation index; $R$ is the number of stress factors; and $K$ is a semi-saturation constant, which is generally $1/2$ of the maximum habitat degradation.

### 3.2.5. NPP

The CASA model was used to calculate the *NPP*. The proportion of photosynthetically active radiation (FPAR) absorbed by plants is a potential driving force for photosynthesis and the growth of any resource. Therefore, the *NPP* can be modeled by the relationship between absorbed photosynthetically active radiation (*APAR*), and the influence of temperature and atmospheric water vapor on *NPP* is also considered in the model [34,35]. Since the NPP can be estimated based on satellite data and ground data, in the CASA model, NPP is the product of APAP and photosynthetic effective utilization ($\varepsilon$) [36]. The calculation formula is as follows:

$$NPP = APAR \times \varepsilon$$

$$APAR = SOL \times FPAR \times 0.5$$

$$FPAR = \min \left[ \frac{SR - SR_{min}}{SR_{max} - SR_{min}}, 0.95 \right]$$

$$SR = \left[ \frac{1 + NDVI}{1 - NDVI} \right]$$

$$\varepsilon = T_{\varepsilon 1} \times T_{\varepsilon 2} \times W_\varepsilon \times \varepsilon_{max}$$

In the formula, NPP, *APAR* and $\varepsilon$ represent the net primary productivity of vegetation (gc·m$^{-2}$), the absorbed photosynthetically active radiation (MJ·m$^{-2}$), and the actual light utilization rate (gc·MJ$^{-1}$), respectively; *SOL* and *FPAR* are the total solar radiation (MJ·m$^{-2}$) and the absorption ratio of the vegetation to photosynthetically active radiation; the constant 0.5 represents the solar radiation rate used by the vegetation; $SR_{min}$ is 1.08, and $SR_{max}$ is related to the vegetation type; *NDVI* represents the vegetation coverage; $T_{\varepsilon 1}$, $T_{\varepsilon 2}$ and $W_\varepsilon$ are the threat effects of low temperature, high temperature, and moisture conditions on the light utilization efficiency, respectively.

### 3.2.6. The Trade-Off and Synergy Analysis

The Spearman rank correlation is a non-parametric method used to measure bivariate relationships. It is often used in trade-off and synergy research [37]. The positive correlation between two ecosystem services indicates that they are synergistic, and the negative correlation is expressed as a trade-off relationship. The calculation formulas are as follows:

$$P_{X,Y} = \frac{cov(X,Y)}{\sigma_X \sigma_Y} = \frac{E((X - \mu_X)(Y - \mu_Y))}{\sigma_X \sigma_Y} = \frac{E(XY) - E(X)E(Y)}{\sqrt{E(X^2) - E^2(X)} \sqrt{E(Y^2) - E^2(Y)}}$$

$$r_g = P_{rgX,rgY} = \frac{cov(rgX, rgY)}{\sigma_{rgX} \sigma_{rgY}}$$

In the formula, $P_{X,Y}$ is the Pearson correlation coefficient of the variables $X$ and $Y$, $cov(X,Y)$ is the covariance of the two variables, $\sigma_X$ and $\sigma_Y$ are the standard deviations of the two variables, and $P_{rgX,rgY}$ is the correlation coefficient of Spearman applied to the rank of the original variable. The R language function software package is used to draw

a correlation coefficient map between ecosystem functions, and to intuitively display the trade-off and synergy relationship between multiple functions.

### 3.2.7. Global Spatial Autocorrelation

In order to further clarify the spatial distribution characteristics of the ecosystem service trade-off and synergy relationship, each ecosystem service was assigned to the vector diagram of the Yellow River basin, and the GeoDa software was used to apply the bivariate local Moran index to the five ecosystem services to complete the bivariate autocorrelation analysis. Moran's I index represents the similarity ratio of unit attribute values in adjacent regions of space. In this paper, the spatial correlation between ecosystem functions of grassland grid units in the Yellow River basin was analyzed by GeoDA. The formulas are as follows:

$$ I = \frac{n \sum_{i=1}^{n} \sum_{j=1}^{n} w_{ij} \left( x_i - \bar{x} \right) \left( x_j - \bar{x} \right)}{\sum_{i=1}^{n} \left( x_i - \bar{x} \right)^2 \left( \sum_i \sum_j w_{ij} \right)} $$

$$ Z(G_i^*) = \frac{\sum_{j=1}^{n} w_{i,j} x_j - X \sum_{j=1}^{n} w_{i,j}}{s \sqrt{\frac{\left[ n \sum_{j=1}^{n} w_{i,j}^2 - \left( \sum_{j=1}^{n} w_{i,j} \right)^2 \right]}{(n-1)}}} $$

$$ X = \frac{1}{n} \sum_{j=1}^{n} x_i, \; S = \sqrt{\frac{1}{n} \sum_{j=1}^{N} x_j^2 - x^2} $$

In the formula, I is Moran's I index, n is spatial cells' number, $x_i$ and $x_j$ are respectively the observed values of the i and j regions, $w_{ij}$ is the spatial adjacency relationship between regions i and j, S2 is the variance of observed values. The Moran's I index was generally between −1 and 1 (0,1) indicating that the geographical entities had a positive correlation. A negative correlation is denoted by (−1,0), and the value of 0 indicates there is no correlation. High-high agglomeration and low-low agglomeration indicate a synergistic relationship, and high-low agglomeration and low-high agglomeration indicate a trade-off relationship.

## 4. Results and Analysis

### 4.1. Temporal and Spatial Characteristics of Grassland Ecosystem Services

The five grassland ecosystem functions were evaluated by adopting the InVEST and CASA models. The results are shown in Table 1. The average water yield depth of grassland in the Yellow River basin in 1990, 1995, 2000, 2005, 2010 and 2018 was in a state of "decrease-increase-decrease-increase". Water yield is mainly affected by rainfall and potential evapotranspiration, and the annual rainfall in 2005 and 2018 was higher than during of other periods, so the water yield depth in the corresponding years was higher. The amount of soil conservation in the grassland showed a continuous decreasing trend, with a total decrease of $1.560 \times 10^8$ t, a decrease of 16.94%. The soil retention of high, medium and low coverage grasslands were $2.185 \times 10^8$ t, $3.970 \times 10^8$ t and $2.419 \times 10^8$ t, respectively. From 1990 to 2018, although the area of grassland in the Yellow River basin increased, the area of high-coverage grassland decreased, and the middle and low-coverage grassland increased, reflecting the trend of grassland degradation in the watershed to a certain extent. In addition to the increase in rainfall, the actual amount of erosion increased. The amount of soil conservation during the study period eventually reduced. During the study period, the grassland's carbon storage showed an increasing trend, with a total increase of $9.044 \times 10^6$ t, an average annual increase of $3.23 \times 10^5$ t, mainly because carbon storage was related to carbon density and grassland area, and the grassland area showed an increasing trend during the study period. According to the classification criteria in the existing research, the habitat quality was divided into 5 levels: low (0–0.2), lower (0.2–0.4), medium (0.4–0.6), higher (0.6–0.8) and high (0.8–1). It can be observed that the grassland

habitat quality of the Yellow River basin is of a relatively high level, and the grassland habitat quality has changed little in the six periods. The grassland NPPs in each stage were 204.66 gc·m$^{-2}$, 190.63 gc·m$^{-2}$, 225.76 gc·m$^{-2}$, 229.73 gc·m$^{-2}$, 255.91 gc·m$^{-2}$ and 296.16 gc·m$^{-2}$, respectively, showing an overall increasing trend.

**Table 1.** Grassland ecosystem functions in the Yellow River basin.

| Year | Water Yield Depth (mm) | Soil Conservation ($10^8$ t) | Carbon Storage ($10^8$ t) | Habitat Quality | NPP (gc·m$^{-2}$) |
|---|---|---|---|---|---|
| 1990 | 42.19 | 9.201 | 26.653 | 0.686 | 204.66 |
| 1995 | 40.62 | 9.098 | 26.176 | 0.688 | 190.63 |
| 2000 | 23.28 | 8.792 | 24.858 | 0.686 | 225.76 |
| 2005 | 69.48 | 8.497 | 25.789 | 0.686 | 229.73 |
| 2010 | 44.40 | 8.201 | 25.351 | 0.686 | 255.91 |
| 2018 | 64.88 | 7.648 | 26.743 | 0.685 | 296.16 |

The water yield is low in the northwest and high in the southwest (Figure 2a). The high water yield areas are mainly concentrated in the basin above Longyangxia, and the low water yield areas are mainly distributed in the Loess Plateau, the Ningxia Plain and the Hetao Plain. Above the Beiluo River and below the right bank of Wubao, this distribution pattern is directly related to the regional average annual precipitation and the distribution pattern of grassland. That is, areas with high average annual precipitation and low vegetation evapotranspiration have a strong water yield capacity. High-value areas and low-value areas of grassland soil conservation have no obvious boundaries; high and low are cross-distributed. The spatial distribution of high values of soil conservation is similar to the distribution of grassland carbon storage, which is mainly distributed in the Ruoergai Plateau, the Qilian Mountains, the Qinling Mountains, and the Luliang Mountains in Shanxi province. On the one hand, because of the large potential erosion in this area and, on the other hand, due to the high vegetation coverage, the grassland types are mostly high-coverage grasslands, and the soil conservation ability is strong. The low-value areas of soil conservation are mainly distributed in the source area of the Yellow River, the western part of the Loess Plateau, and the central area of the Loess Plateau (Figure 2b). The high-value areas and low-value areas of grassland carbon storage are distributed throughout the basin. The high-value areas account for a large area and are more concentrated in space. They are mainly distributed in the upper Ruoergai grassland, the lower Xiaolangdi, Huayuankou, and the Weihe Valley. Scattered in various areas in the middle reaches, this is a concentrated distribution area with a high level of grassland coverage across the whole watershed, with high carbon density; while low-value areas are mainly distributed in the middle of the Loess Plateau and the transitional area between the Loess Plateau and the Qinghai-Tibet Plateau, where there are mainly low-coverage grasslands with low carbon density (Figure 2c). The high-value areas of grassland habitat in the Yellow River basin are mainly concentrated in the Qinghai-Tibet Plateau, where the source area of the Yellow River is located, and scattered in the Mu Us Sandy Land, the Qinling Mountains, and the southern part of the Loess Plateau. The low-value areas of habitat quality are mainly concentrated in the lower reaches of the Yellow River basin, the Guanzhong Plain, and the Fenhe Valley (Figure 2d). The high-value areas of grassland NPP in the Yellow River basin are mainly distributed in the Qinghai-Tibet Plateau, where the source area of the Yellow River is located, and scattered in the Mu Us Sandy Land, the Qinling Mountains, and the southern part of the Loess Plateau. The low-value areas are mainly concentrated in the source area of the Yellow River in the Yellow River basin, the western part of the Loess Plateau region, and the central region of the Loess Plateau (Figure 2e).

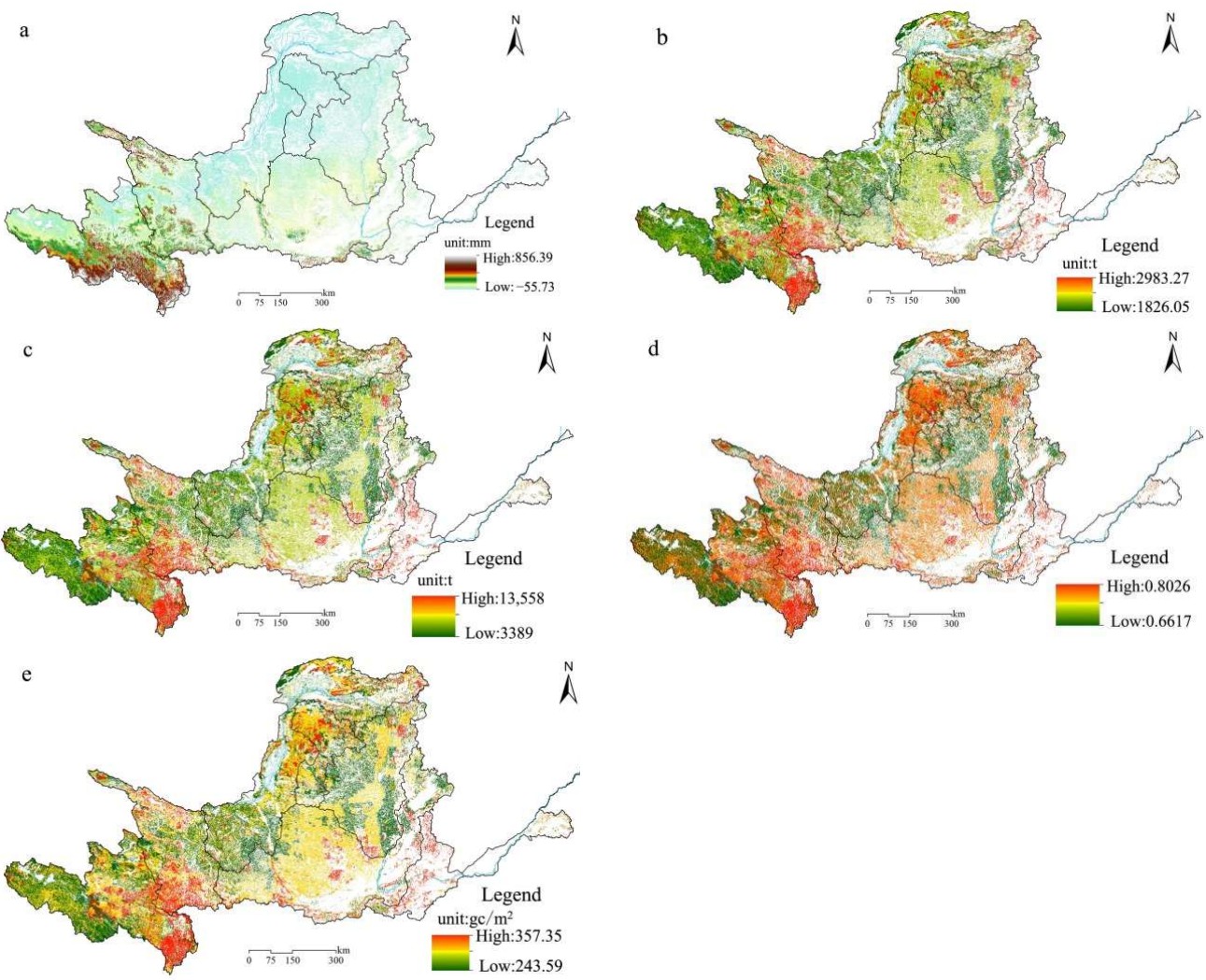

**Figure 2.** Spatial distribution of five ecosystem services in the grassland of the Yellow River basin ((**a**): water yield, (**b**): soil conservation, (**c**): carbon stock, (**d**): habitat quality, (**e**): NPP).

### 4.2. Changes in Trade-Offs and Synergistic Relationships between Grassland Ecosystem Functions from 1990 to 2018

From Figure 3, it can be seen that the relationship between various grassland services from 1990 to 2018 showed a more consistent law, and the trade-off and synergy relationships between various functions were relatively stable. The positive correlation coefficient between water yield and other services was very low, showing a weak synergistic relationship, while the correlation coefficients between the other services were larger; all were greater than 0.5, showing strong synergy. Although the overall trade-off and synergy relationship did not change, the correlation coefficient between the grassland ecosystem functions fluctuated. From the perspective of correlation coefficients, the correlation coefficients between water yield and NPP, habitat quality, carbon storage, and soil conservation changed little during the study period, and showed a slight upward trend. The correlation coefficients between the other groups of functions all showed a decreasing trend. The correlation coefficient between the NPP and the soil conservation declined the most, with a decrease of 0.35, followed by carbon storage and soil conservation, with a decrease of 0.24.

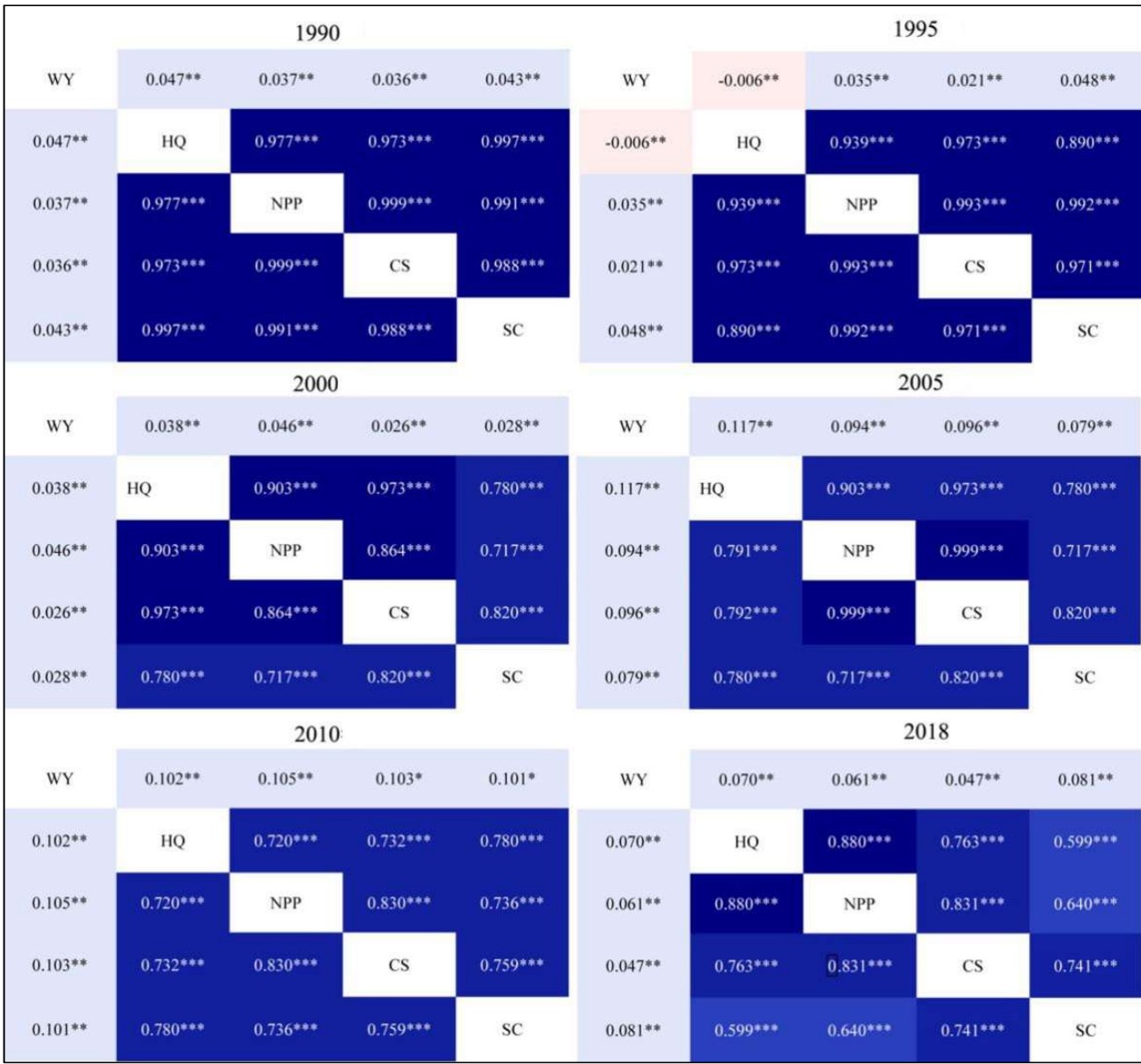

**Figure 3.** Trade-off and synergy coefficient of five ecosystem services of grassland in the Yellow River basin from 1990 to 2018 (WY: water yield; CS: carbon storage; HQ: habitat quality; SC: soil conservation). Note: *** $p < 0.01$; ** $p < 0.05$; * $p < 0.1$.

### 4.3. Spatial Expression of Trade-Offs and Synergistic Relationships between Grassland Ecosystem Functiond

In order to express the spatial trade-off and synergy of the grassland ecosystem functions in the watershed more intuitively, GeoDA software was applied to conduct a bivariate local spatial autocorrelation analysis. The result was that high-high agglomeration and low-low agglomeration indicated a positive correlation as a synergy relationship. However, agglomeration with high-low and low-high indicated that the negative correlation is a trade-off relationship (Figure 4). Water yield and soil conservation (WY-SC), water yield and NPP (WY-NPP), and the high and high-agglomeration areas (high-value synergy) were mainly distributed in the Gannan Plateau, the Qilian Mountains and Qinling areas, and the low and low-agglomeration areas (low-value synergy areas) were mainly distributed in the Yinchuan Plain, the Hetao Plain, the Mu Us Sandy Land and the lower reaches of the Yellow River basin. The above-mentioned areas are synergistic areas for water yield and soil conservation. In the Taihang Mountain area, water yield and soil conservation

were in a trade-off relationship, with low water yield and high soil conservation. The synergy areas between water yield and habitat quality (WY-HQ) and water yield and carbon storage (WY-CS) were more consistent. The high-value agglomeration areas of the two were mainly distributed in the Zoige grassland, Gannan Plateau and Datong River basin. The above-mentioned areas are grasslands, which experience abundant rainfall and high water yield. At the same time, due to the good vegetation conditions in this area, its habitat quality and carbon storage are both high, showing a high-value synergy zone. Low-value agglomeration areas were different. They were mainly distributed in the lower reaches of the Yellow River. The common low-value areas of water yield and carbon storage were relatively scattered, concentrated in internal flow areas, west of the Loess Plateau and the lower reaches of the Yellow River. The trade-off areas (low-high and high-low) were mainly distributed in the Fenhe River basin. The overall performance pointed to low water yield and high habitat quality and carbon storage. The synergy areas between soil conservation and NPP (SC-NPP) were similar to those between carbon storage and NPP (CS-NPP) (high-value agglomeration areas and low-value agglomeration areas). The high-value agglomeration areas were mainly distributed in Zoige grassland, the Datong River basin, the Qinling Mountains and areas in the Fen River valley. The low-value areas were mainly distributed in the source area of the Yellow River, the transitional area between the Qinghai-Tibet Plateau and the Loess Plateau, and most of the Loess Plateau, indicating that soil conservation, carbon storage and NPP were generally synergistic. Soil conservation and habitat quality (SC-HQ) showed a large-scale trade-off relationship in space, mainly distributed in the Inner River, the Yinchuan Plain, and the Hetao Plain, with high habitat quality and low soil conservation. The area is mainly unused land, and there are few threat sources, such as construction land and cultivated land, that threaten the quality of the habitat; the habitat quality is good. Meanwhile, the vegetation coverage is low, and the soil conservation is weak. The synergistic regions of carbon storage and habitat quality (CS-HQ) were mainly distributed in Zoige grassland, the Gannan Plateau, the Qinling Mountains and the Weihe Valley, and the low-value regions were distributed in the lower reaches of the Yellow River basin. The high-value areas of grassland carbon storage and soil conservation (CS-SC) were distributed in Zoige grassland, the Datong River basin, the Qinling area and the valley areas in the Fen River, and the low-value areas were mainly distributed in Liupan Mountain area, inland and watershed, and downstream areas.

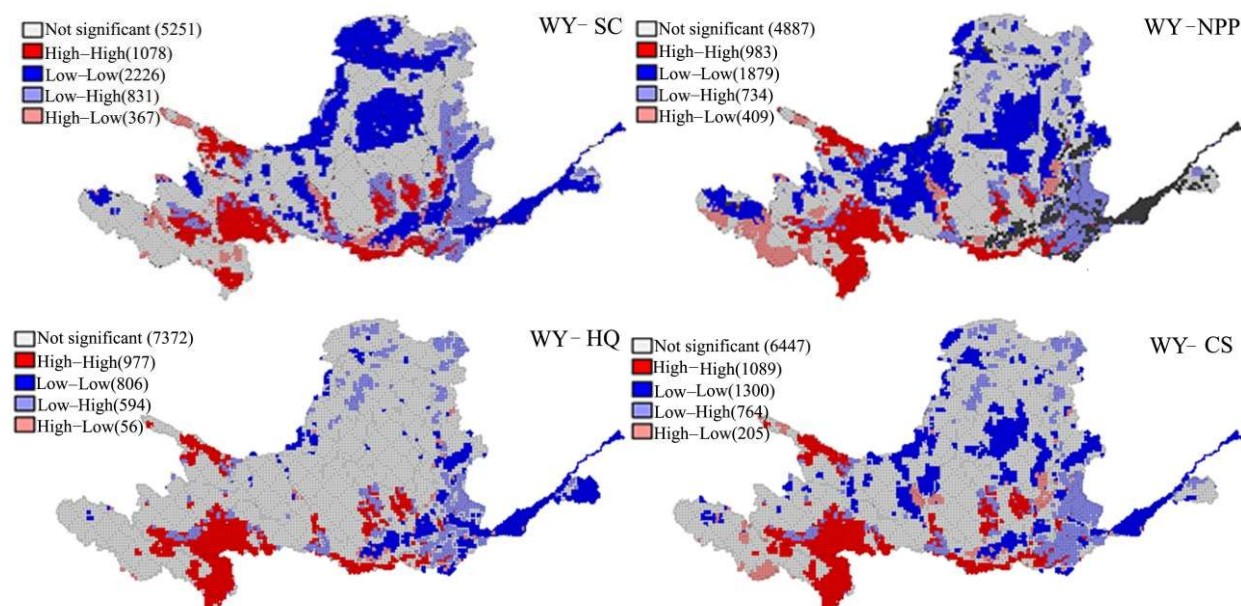

**Figure 4.** *Cont.*

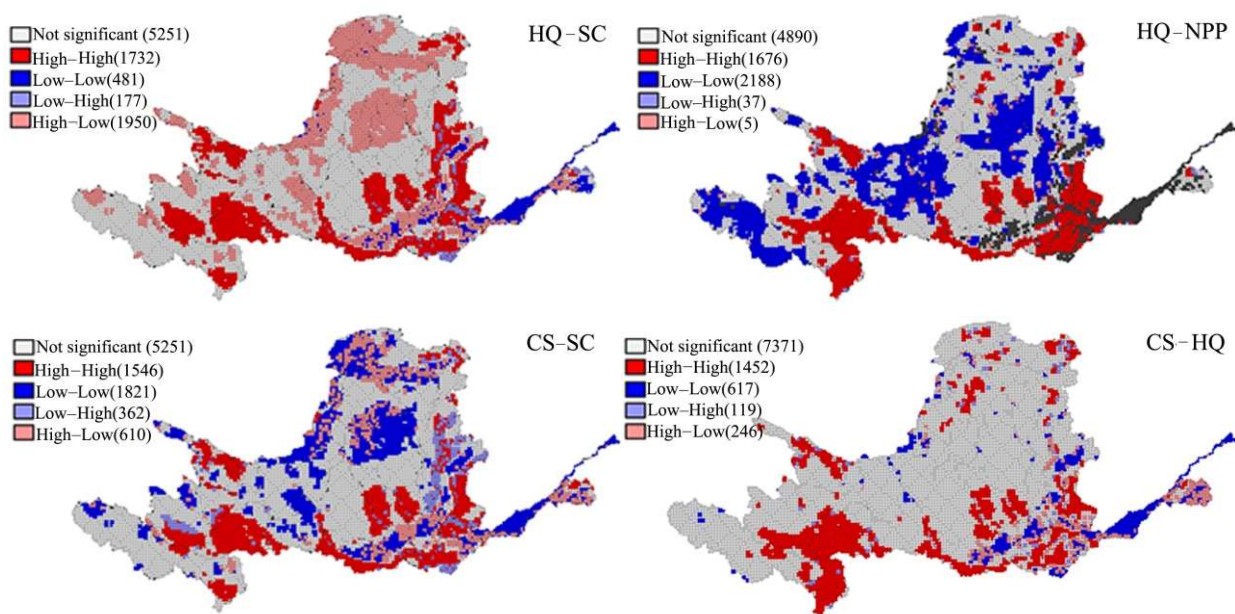

**Figure 4.** Bivariate spatial correlation analysis of ecosystem service of grassland system. (WY: water yield; CS: Carbon Storage; HQ: Habitat Quality; SC: Soil Conservation).

## 5. Discussion

Identifying the interrelationships between ecological functions is essential for understanding services and guiding decision-making However, the interactions between ecological functions are not all linear. Therefore, the interaction between them can be understood only by evaluating multiple functions [38]. This study selected five typical grassland ecological services, analyzed their trade-off sand synergistic relationships by adopting the correlation coefficient method and the bivariate spatial autocorrelation method, explored the temporal change characteristics of the trade-offs and synergistic relationships, and expressed the heterogeneity in space. The relationships between the five typical ecological functions of grasses in the Yellow River basin were relatively stable. Except for the weak trade-off relationship between water yield and habitat quality in 1995, all the functions in the other years were synergistic, showing that the five ecological services, water yield, carbon storage, soil conservation, habitat quality, and NPP are mutually beneficial. This is because, with the increase in water yield, the grassland area and grassland coverage increase overall. Studies have revealed that the areas of grassland in the Yellow River basin are increasing due to the implementation of the National Returning Farmland to Forest (Grass) Project, which has clearly resulted in ecological restoration in the Yellow River basin [39], including increases in carbon storage. At the same time, grassland has the ability to intercept rainfall and consolidate soil, which has reduced soil loss, and has strengthened the soil's conservation capabilities [40]. With the overall improvement in grassland coverage, the quality of its habitat and NPP will inevitably increase. The synergy between water yield and habitat quality was the strongest measured in the study area, although it showed a weak trade-off in 1995, indicating that there are temporal differences in the relationships between ecosystem functions. A more in-depth analysis of the long-term changes in trade-offs and synergistic relationships in the study area and its spatial relationship characteristics should be carried out in the future [41].

There were certain similarities and differences between the trade-off and synergy results of the five service functions of the grassland ecosystem in the Yellow River basin and related studies. Supply and other services usually present a trade-off relationship, especially between water yield services and regulation services, such as carbon fixation, pollutant retention, and soil conservation [42]. The results of this study show that grassland water yield had a synergistic relationship with carbon storage, soil conservation, habitat

quality, and NPP, which is consistent with the results of Qian et al. [43]. The relationships between the other functions were shown to be synergistic. From the perspective of the driving factors of the trade-offs and synergy of grassland ecosystems, slope and NDVI had significant effects on soil conservation, carbon storage, habitat quality, and NPP. This was because the common driving factors led to a synergistic relationship of increase and decrease between them, while the water yield was mainly affected by rainfall and slope. Therefore, there was weak synergy between the water yield and the other four functions, and the degree of synergy between options and balances was shown to have changed from 1990 to 2018. The external form and internal structure of the ecosystem are constantly changing, and this dynamic change makes different scales display different research results [44]. The function of grassland ecosystems are derived from the interaction between landscape patterns (patch structure, corridor layout, landscape diversity, etc.), ecological processes (regional water and soil process and global climate change), and the scale constraint of the effect; that is, ecosystem functions have spatial heterogeneity, and the direct interaction between system services or the temporal and spatial differences between common driving factors also lead to changes in the trade-offs and synergy between services [45]. The Yellow River basin is wide in scope, with a large span from east to west, and north to south, and it is a transition zone between eastern and western China. Transition has an impact on regional barriers and differentiation, resulting in differences in topography, landforms, and climate. The types of grassland ecosystems are also different, which has an impact on grassland ecosystem functions and creates spatial differences between the grassland ecosystem's functions.

## 6. Conclusions

From 1990 to 2018, the average water yield depth, carbon storage, and NPP of the grassland showed an increasing trend, while the soil conservation and habitat quality of the grassland showed a downward trend. The five grassland ecosystem functions demonstrated different spatial distributions, with water yield low in the northwest and high in the southwest; the spatial distribution of carbon storage and soil conservation was similar, and the habitat quality was more consistent with the spatial distribution of the NPP.

From 1990 to 2018, the relationships between various grassland functions were relatively consistent. The positive correlation coefficient between water yield and other functions was low, showing a weak synergistic relationship, while the correlation coefficients among the other services were relatively high.

The trade-offs and synergistic relationships between various grassland functions displayed regional heterogeneity in space, and the relationships between some services were spatially inconsistent with the results of the time series, indicating that the trade-offs and synergistic relationships between grassland ecosystem functions featured clear differences in scale.

**Author Contributions:** Conceptualization, J.Y., B.X. and D.Z.; Identification, J.Y.; Writing—review and editing, J.Y., B.X. and D.Z.; Project administration, J.Y.; Funding administration, B.X. All authors have read and agreed to the published version of the manuscript.

**Funding:** This research was was funded by the Scientific Research Start-up Funds for Openly Recruited Doctors (GAU-KYQD-2017-34).

**Institutional Review Board Statement:** Not applicable.

**Informed Consent Statement:** Not applicable.

**Data Availability Statement:** The processed data required to reproduce these findings cannot be shared at this time as the data also forms part of an ongoing study.

**Conflicts of Interest:** The authors declare no conflict of interest.

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
