# Peer review of "The Trade-Offs and Synergistic Relationships between Grassland Ecosystem Functions in the Yellow River Basin"

_diversity, doi:10.3390/d13100505_

Round 1

Reviewer 1 Report

Yang et al. examined the trade-offs and synergies of grassland ecosystem services along the Yellow River Basin between 1990 and 2018. They specifically explored the temporal trends of five grassland ecosystem services, including water production, carbon storage, soil conservation, habitat quality, and NPP. They used the correlation coefficient method to investigate the trade-offs and synergies among the five ecosystem services and used the bivariate spatial autocorrelation method to investigate the spatial heterogeneity of trade-offs and synergies of the five grassland ecosystem services. They found that ecosystem services generally increased across the study period while soil conservation and habitat quality decreased to some extent. All services showed synergistic relationships that varied across space. This work presents a novel way to study the temporal- and spatial-dynamics in grassland ecosystem services and showcases the synergies of ecosystem services at the regional scale. This work may contribute to the assessment of regional and local ecosystem services.

I have some major issues as follows:

  1. The wording, trade-off, consistently presents throughout the manuscript. It could be misleading because the authors found synergistic relationships among all the five ecosystem services considered. Although weak correlations were observed between water production and the other four ecosystem services, trade-offs were not quite often observed. I understand that the authors introduced trade-offs in the Introduction section, but it’s vague to use the wording, trade-off, in the Title, Abstract and Results sections. In addition, I understand that the authors conducted a bivariate autocorrelation analysis, and considered high-low/low-high glomeration as a trade-off relationship. However, I don’t understand the differences between the results of correlation analysis (all synergistic) and the results of bivariate autocorrelation analysis (some synergistic, while some trade-off).
  2. The authors take great efforts to discuss the influences of driving factors on the balance of synergies. I understand grazing and human activities may affect the relationships between ecosystem services. However, the authors did not conduct a driving factor analysis to quantify the relative importance of economic development, climate change, grazing, and other human activities on the synergies of ecosystem services. I, therefore, worry that the authors overestimated some aspects of the driving factors, while others were underestimated across the whole Yellow River Basin. For example, grazing is an important driving factor in the upstream regions of the Yellow River Basin, while it may become less important in the downstream regions of the Yellow River Basin. It, therefore, seems to be reasonable to conduct the driving factor analysis across different spatial scales.

Minor comments (Note that I may have missed the version with line numbers, but I did not find line numbers and will not use it below. In addition, I focused on the scientific aspect of the manuscript, and did not check the typos and grammar issues):

Title

  1. See my comments above on the trade-off and synergistic relationships.
  2. replace grassland ecosystem with grassland ecosystem services

Abstract

  1. Do you mean high-quality economic development?
  2. “This paper quantitatively evaluates the five typical services of grassland in the Yellow River Basin, including water production, carbon storage, soil conservation, habitat quality and NPP, adopting the InVEST model and the CASA model, analyzes the changes of the trade-off and synergistic relationship among the five ecosystem services from 1990 to 2018, adopting the correlation coefficient method, and analyzes and explores spatial heterogeneity of the trade-off and synergistic relationship, adopting the bivariate spatial autocorrelation method.” This is a long sentence, please convert it into short sentences.
  3. Please add a conclusion sentence at the end of the abstract.

Introduction

  1. Please elaborate on the point that “the core reason for the above problem is the structural imbalance of various ecological services of grassland, such as overgrazing”. In addition, I don’t understand the meaning of “structural imbalance of various ecological services of grassland”.
  2. I cannot follow the meaning of “their mechanism of action and changes”.
  3. Please insert references to support “The spatial heterogeneity and time dependence of the trade-off and synergy have been verified by relevant researches”.
  4. Please don’t use the given name when cited a paper in the second paragraph of the Introduction, e.g., refs. 8, 9, 12.
  5. The Year of publication was missing for Zhong et al.
  6. I don’t understand why the methods used to quantify trade-offs were introduced. The sentence does not fit in the context. “The existing studies on the trade-off and synergy relationship of ecosystem services mostly use correlation coefficient method [17], principal component analysis method [18], cluster analysis method [19], production possibility boundary method [20], rose diagram [21] and other methods.”
  7. Please replace our country with China throughout the manuscript.
  8. Research questions and working hypotheses were missing in the last paragraph of the Introduction.

Methods

  1. It’s unclear whether Spearman rank correction or Pearson correlation was used in this work because both were mentioned in the section of “the trade-off and synergy analysis”.
  2. I am not familiar with the bivariate autocorrelation analysis. I was wondering if you could provide more details on the analysis.

Results

  1. Can you replace water yield with water production?
  2. Please use the full name of ecosystem services if possible in Figures 3 and 4.

Discussion

  1. What does it mean for “a correct understand of the trade-off and synergy”?
  2. I don’t understand what “the ecological project” represents.

Author Response

Response to Reviewer 1 Comments

Thank you for the reviewers’ comments concerning our manuscript entitled “The Trade-off and Synergistic Relationship of Grassland Eco-system in the Yellow River Basin” (manuscript number: diversity-1403490). Those comments are all valuable and very helpful for revising and improving our paper, as well as important guiding significance to our researches. We have studied the comments carefully and have made correction. We really hope to meet with your approval. The revised portions are marked in red in the paper. The main corrections in the paper and the responses to the reviewer’s comments are as follows:

Point1:  The wording, trade-off, consistently presents throughout the manuscript. It could be misleading because the authors found synergistic relationships among all the five ecosystem services considered. Although weak correlations were observed between water production and the other four ecosystem services, trade-offs were not quite often observed. I understand that the authors introduced trade-offs in the Introduction section, but it’s vague to use the wording, trade-off, in the Title, Abstract and Results sections. In addition, I understand that the authors conducted a bivariate autocorrelation analysis, and considered high-low/low-high glomeration as a trade-off relationship. However, I don’t understand the differences between the results of correlation analysis (all synergistic) and the results of bivariate autocorrelation analysis (some synergistic, while some trade-off).

Response1: What needs to be explained is that ecosystem services have strong spatial dependence. This is because the external form and internal structure of ecosystems are constantly changing. This dynamic change makes different scales have different research results.The difference between the results of correlation analysis and bivariate autocorrelation analysis is due to the different scales of the research, which just indicates that the ecosystem service relationship has obvious scale effects, which is also one of the significance of this research.

Point2: The authors take great efforts to discuss the influences of driving factors on the balance of synergies. I understand grazing and human activities may affect the relationships between ecosystem services. However, the authors did not conduct a driving factor analysis to quantify the relative importance of economic development, climate change, grazing, and other human activities on the synergies of ecosystem services.  therefore, worry that the authors overestimated some aspects of the driving factors, while others were underestimated across the whole Yellow River Basin. For example, grazing is an important driving factor in the upstream regions of the Yellow River Basin, while it may become less important in the downstream regions of the Yellow River Basin. It, therefore, seems to be reasonable to conduct the driving factor analysis across different spatial scales.

Response2: In the revised manuscript, a discussion of this issue was added to try to explain the reason for the discrepancy.We have modified the original manuscript. On the one hand, there is no in-depth study of the driving factors that affect the balance of synergy, which should be a harder job, and it’s not scientific to do a simple discussion in the discussion parts. On the other hand, this research mainly focuses on weighing the spatial dependence of trade-off and synergy relationship. Therefore, in the revised manuscript, the reasons for the spacial difference of the trade-off and synergy relationship has been added in the discussion part.

Point3: replace grassland ecosystem with grassland ecosystem services

Response3: Thank you for your valuable suggestions and references. We have revised the Title of the manuscript according to your comments.

Point4: About Abstract

Response4: We have revised the Abstract of the manuscript according to these references.

In 2019, the Chinese government put forward the strategy of ecological conservation and high-quality development of the Yellow River Basin.

Point5: About Introduction.

Response5: We have revised the Introduction of the manuscript according to your comments.

Point6: About Methods.

Response6: In the manuscript, we add 3.2.7 to explain the spatial autocorrelation theory in detail.

Point7: About Results.

Response7: In the manuscript, we unify the manuscript into Water Yield

Point8: About Discussion

Response7: We have re-written this part according to the Reviewer's suggestion.

Reviewer 2 Report

Dear Authors,

I enjoyed reading your paper and the research is interesting. Just a minor correction:

Under section 4.1, lines 4-6, you stated that '..the annual rainfall in 1995 and 2018 id higher than that of other periods,..'. Table 1 shows otherwise. It was 2005 and 2018 that had the highest water depths. Was this an error? if so, change '1995' in your text to '2005'.

Author Response

Response to Reviewer 2 Comments

Thank you for the reviewers’ comments concerning our manuscript entitled “The Trade-off and Synergistic Relationship of Grassland Eco-system in the Yellow River Basin” (manuscript number: diversity-1403490). Those comments are all valuable and very helpful for revising and improving our paper, as well as important guiding significance to our researches. We have studied the comments carefully and have made correction. We really hope to meet with your approval. The revised portions are marked in red in the paper. The main corrections in the paper and the responses to the reviewer’s comments are as follows:

Point:  Under section 4.1, lines 4-6, you stated that '..the annual rainfall in 1995 and 2018 id higher than that of other periods,..'. Table 1 shows otherwise. It was 2005 and 2018 that had the highest water depths. Was this an error? if so, change '1995' in your text to '2005'.

Response: Thank you for your valuable suggestions and references. We have revised the Title of the manuscript according to your comments.

Round 2

Reviewer 1 Report

My main concerns have been well addressed. I have no further comments.